# Chemically routed interpore molecular diffusion in metal-organic framework thin films

Tanmoy Maity ®[1,2], Pratibha Malik ®[1,2], Sumit Bawari[1], Soumya Ghosh[1], Jagannath Mondal[1] & Ritesh Haldar ®[1] ✉

Transport diffusivity of molecules in a porous solid is constricted by the rate at which molecules move from one pore to the other, along the concentration gradient, i.e. by following Fickian diffusion. In heterogeneous porous materials, i.e. in the presence of pores of different sizes and chemical environments, diffusion rate and directionality remain tricky to estimate and adjust. In such a porous system, we have realized that molecular diffusion direction can be orthogonal to the concentration gradient. To experimentally determine this complex diffusion rate dependency and get insight of the microscopic diffusion pathway, we have designed a model nanoporous structure, metal-organic framework (MOF). In this model two chemically and geometrically distinct pore windows are spatially oriented by an epitaxial, layer-by-layer growth method. The specific design of the nanoporous channels and quantitative mass uptake rate measurements have indicated that the mass uptake is governed by the interpore diffusion along the direction orthogonal to the concentration gradient. This revelation allows chemically carving the nanopores, and accelerating the interpore diffusion and kinetic diffusion selectivity.

Molecular diffusion in a nanoporous solid, e.g., zeolite, porous carbon, metal-organic framework (MOF) and covalent organic framework (COF)[1–5], is an important process with regard to chemical separation[6,7] and catalysis[8,9]. For separating chemicals, state-of-the art nanoporous membranes[6] require faster diffusion or permeation of the separated chemicals across the membrane layer, so that the production efficacy increases and cost is reduced. In heterogeneous catalysis using the nanoporous solids, reactant diffusion to the active site is the rate-determining step[8,10–12]. Hence, for both of the applications, efficacy of the process is controlled by the diffusivity ($D$). In the case of perfect molecular sieving (i.e., size-based exclusion)[13,14], exclusively selective molecular diffusion occurs while in the case of competitive diffusion of the molecules in the pores, selectivity is decreased. However, the selectivity can be improved by specific pore environment design at different length-scales; few Ångstrom to nanometer-sized pores can be

geometrically and chemically tuned or the nanoporous channels can be oriented in a specific direction at micron scale to accelerate diffusion[15–21]. To formulate these strategies that can accelerate diffusion and consequently the selectivity, insight into the rate-determining step is necessary.

In the nanoporous materials, following physical processes take place during the permeation of the chemicals along the concentration gradient: (A) adsorbate-pore surface interaction, (B) surface to pore diffusion and (C) interpore diffusion. The surface barrier phenomenon[22–25] (i.e., transport resistance due to structural defects and pore blocking) is related to the steps A and B. In certain cases, in particular for MOF thin films, surface barriers influence the diffusion rate. Surface barriers are omnipresent; however it can be substantially minimized by changing the synthetic conditions[22,26]. In case of vanishing surface barrier effect, step C is the rate limiting factor[6,27]. As the

[1]Tata Institute of Fundamental Research Hyderabad, Gopanpally, Hyderabad 500046 Telangana, India. [2]These authors contributed equally: Tanmoy Maity, Pratibha Malik. ✉e-mail: riteshhaldar@tifrh.res.in

permeation is directly proportional to the $D$ and adsorbate solubility, managing the interpore diffusion (step C) is the key in case of nanoporous solids. Earlier studies revealed that the diffusion in the nanoporous solids, e.g., MOFs, can be modeled and estimated using the Fick's law[28–30]. However, in the case of nonlinearity in diffusion (i.e., diffusivity as a function of mass loading in step C), an appropriate model is difficult to formulate[29,30]. This nonlinearity increases with increasing mass loading, as adsorbate-adsorbate interaction also comes into play[31]. Further, in the case of the nonhomogeneous pores (i.e., more than one types of pore window sizes and functionalities)[29,32], which is commonly the case for nanoporous MOFs and COFs, estimation of $D$ also remains tricky. In this communication we postulate that in the absence of substantial surface resistance, the interpore diffusion can be controlled using a chemically derived path at the nonlinear regime of mass loading.

The exact estimation of the molecular diffusion path (and tortuosity)[33] in a nanoporous solid may not be straight forward in the presence of structural defects and disordered crystalline domains[30]. Molecular simulations can be useful to understand the complex, pore topology-dependent diffusion characteristics[34–37]. By experimental route, it is rather more useful to assess the factors that control the interpore diffusion and find out a convenient way to tune those factors. One way to do so is to make a model porous structure and carefully analyze the mass uptake rate ($M_{rate}$). As a proof of concept, we have chosen a nanoporous system in which the pores are highly ordered; one type of the pore windows is aligned along the concentration gradient and another type is orthogonal to the gradient (Fig. 1). The oriented windows are created by metal-organic ligand coordination in a layer-by-layer (lbl) liquid-phase epitaxy (LPE) method, i.e., surface-anchored MOF thin films[38] and solvent vapors are used to probe the $M_{rate}$. Presence of the two chemically and geometrically distinct windows that are perfectly aligned orthogonal to each

other helps to realize the interpore diffusion directionalities. It is revealed that $D$ is not controlled by surface barriers and the interpore diffusion in the selected structure is actually controlled by the orthogonally positioned (to the concentration gradient) pore windows, but not those which are aligned along the concentration gradient. These findings assist to tune the molecular diffusion process using a chemically derived route. We have adopted an isoreticular MOF design strategy[39] to introduce different chemical functionalities in two isostructural MOFs, and in the following discussion we demonstrate its impact on the molecular diffusivities with supporting mass uptake rate experiments and simulations.

## Results

While considering a model porous structure, we have set the following criteria: (i) preconceivable nanometer pore size and geometry, (ii) periodically arranged pores with specific orientation, (iii) chemical tunability, and (iv) ease of assembly as a thin film at micrometer length scale (so that it can be related to a membrane-type structure). Among the contemporary porous materials, MOFs qualify with these criteria. MOFs consist of inorganic metal or metaloxo nodes and functionalized organic linkers[40,41], which are linked by reversible and directional coordination bonds. The choices for metal and linkers are virtually infinite, and the possible structural topologies are also numerous. To name a few benchmark examples where molecular diffusion and gas adsorption selectivities have been studied in details and possible applications for membrane-based gas separation have also been performed, are ZIFs (Zeolitic imidazolate frameworks), UiOs (University of Oslo), MILs (Material Institute Lavoisier)[38,42–46]. In our present approach, we have considered a rather simple PCU topology that can afford two different pore windows. One advantage of this type of topology, otherwise also known as pillared-layer MOFs[47–49], is that these structures can be grown as a thin film in an oriented fashion[50–52] and two different types of pore windows can be arranged in a preconceived orientation.

The selected model structure is Cu(BDC)(pillar) MOF, where the pillars are Py-X = X-Py (Py = pyridyl, X = CH and N) (Fig. 2a). The Cu(BDC) 2D square grids are formed by linking Cu-paddle-wheels with benzenedicarboxylic acid (BDC) linker along the $ab$ plane and these 2D sheets are pillared by Py-X = X-Py along the $c$-axis (along [001]) forming an extended pillared-layer structure (Fig. 2a). This 3D structure features two types of pore windows, one of them has a size of ~7.3 × 4.3 Å along the $c$-axis while the other one is ~9.7 × 6.9 Å along the $ab$ plane. These window sizes are estimated by adding van der Waals radii of the atoms in the simulated structures (see computational details). Herein, we have two types of pillared-layer (PL) structures, denoted as $PL_{C=C}$ and $PL_{N=N}$. The only difference between the two structures is their pillar linker functionality, one having -C = C- while the other one with -N = N-. Note that the smaller pore windows are chemically equivalent but different chemical functionalities are present at the larger pore windows (~3-times larger, Fig. 2a).

To synthesize the model structures and perform the molecular diffusion studies, we have grown oriented thin films of both $PL_{C=C}$ and $PL_{N=N}$ using well-known LPE method in an lbl fashion (see experimental section). By repeating the number of deposition cycles we could obtain homogenous and pinhole free ~200 nm thick films (Fig. 2a, see Supplementary Fig. 1). These synthesized films were characterized using powder X-ray diffraction (PXRD) and Raman spectroscopy (Supplementary Figs. 2 and 3, respectively). Figure 2b shows the out-of-plane (OP) PXRD along with the simulated PXRD patterns. In the OP PXRD, the diffraction peaks appear at ~5.4, 10.8, and 16.3°. Comparison of these peaks with simulated PXRD suggests that these peaks are related to (00l) planes of the $PL_{C=C}$. In the in-plane PXRD, we have observed the diffraction peaks corresponding to the orthogonal planes ((100), (010) and (110), see Supplementary Fig. 2). This observation confirms that the $PL_{C=C}$ structure is oriented

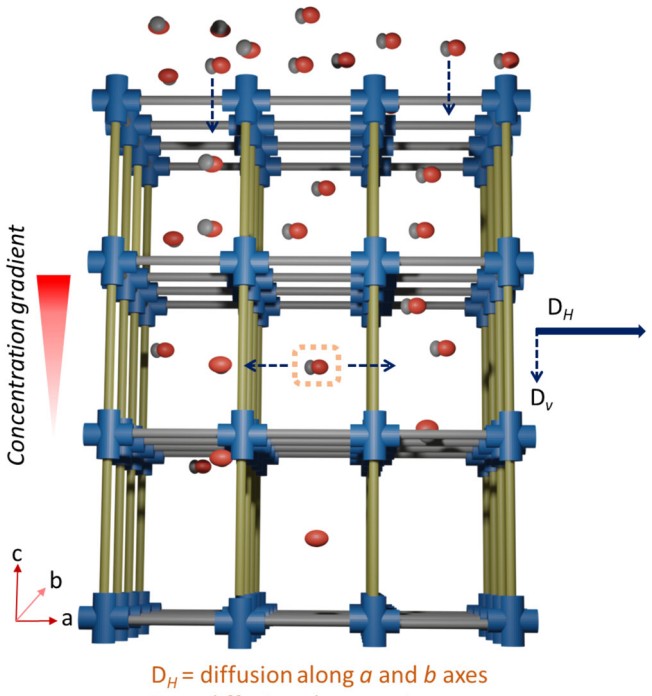

$D_H$ = diffusion along *a* and *b* axes
$D_V$ = diffusion along *c* axis

**Fig. 1 | Molecular diffusion in MOF.** An illustration of the molecular diffusion along the concentration gradient in a crystalline, porous structure, in which mass uptake is controlled by the pore windows orthogonal to the concentration gradient; (molecules are methanol, presented in space fill model, O = red, C = gray); concentration gradient is along *c*-axis. Color code: blue Cu-paddle-wheel, gray BDC linker, yellow pillar linker.

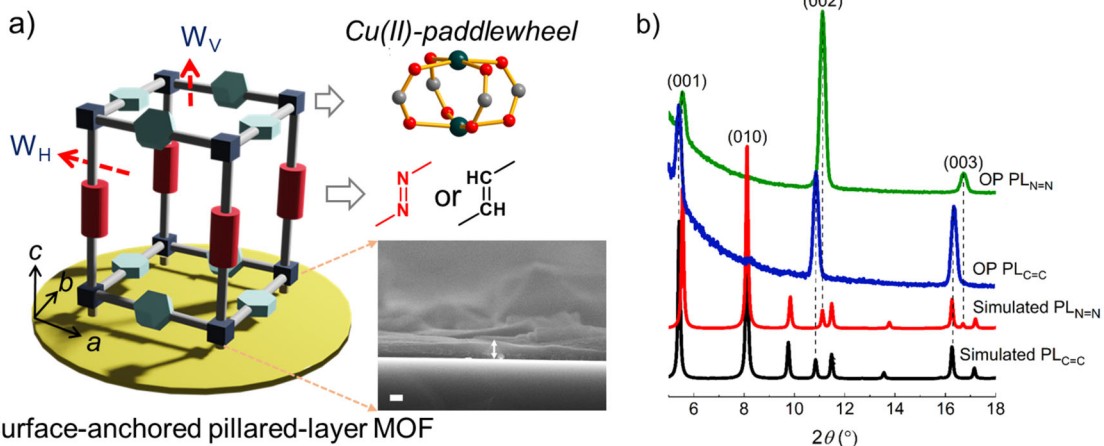

**Fig. 2 | Design of the pillar-layered MOF structure.** Structural insight of the model nanoporous MOF structure: **a** a pillared-layer surface-anchored MOF, with two distinct pore windows $W_V$ and $W_H$; inset figures illustrate the chemical constituents of the MOF and scanning electron microscopy image of the $PL_{C=C}$ MOF (scale bar = 100 nm), **b** comparison of the simulated and out-of-plane XRD patterns of the PL thin films. Color code: Cu = green, O = red, C = gray.

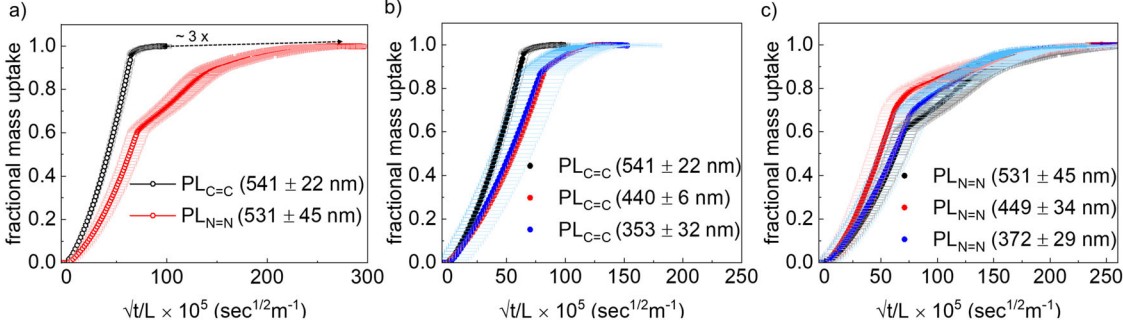

**Fig. 3 | Methanol vapor uptake rates.** Mass uptake rate studies: **a** fractional methanol vapor mass uptake rate profiles at 298 K for $PL_{C=C}$ and $PL_{N=N}$, **b**, **c** fractional methanol vapor uptake rate profiles with different film thickness for $PL_{C=C}$ and $PL_{N=N}$, respectively. Error bars calculated using thickness of the films.

along the (001) or *c*-direction where the smaller pore windows are vertically aligned, (hereafter called as $W_V$) and the larger windows are parallel to the substrate plane (along the *ab* plane, hereafter called as $W_H$). $PL_{N=N}$ thin film also exhibits similar growth orientation, as can be confirmed from the PXRD patterns (Fig. 2b and Supplementary Fig. 2).

Because of the crystal growth preference along the *c*-axis, the surfaces of both thin films are populated with the $W_V$ windows as shown in Fig. 2a. Hence, during the molecular diffusion into the thin film steps A and B (*vide supra*) should be similar for both $PL_{C=C}$ and $PL_{N=N}$. Diffusivity will differ, only if the different chemical functionalities come into play or the larger pore windows $W_H$ control the diffusivity. To study this, we have measured mass uptake rates of the PL thin films grown on quartz crystal microbalance (QCM)[28] sensors with –OH functionalized Au-surface. Methanol (kinetic diameter ~3.6 Å)[53] is used as a probe molecule because it is compatible with the pore window sizes and has high vapor pressure at ambient temperature. The QCM sensors coated with the PL thin films were mounted in a fluidic cell in a temperature-controlled environment. The saturated methanol vapor (~16.8 kPa)[54] uptake profiles were recorded at 298 K by monitoring the fundamental frequency change ($\Delta f$) over time (t). The mass change ($\Delta m$) per area is calculated using the Sauerbrey equation:

$$\Delta m = -c\frac{\Delta f}{n} \quad (1)$$

where *n* denotes the overtone order and *c* is the mass sensitivity constant[28].

In Fig. 3a, the fractional mass uptake is plotted against the uptake time in linear and logarithmic scale. At lower fractional uptake (<20%; molecules entering from the vapor phase into the pore, i.e., steps A and B) both $PL_{C=C}$ and $PL_{N=N}$ shows linear uptake behavior and almost no difference in the uptake rate ($D = 5.7 \times 10^{-16} \pm 1.2$ for $PL_{C=C}$ and $2.4 \times 10^{-16} \pm 0.8$ m²/s for $PL_{N=N}$ at lower % uptake). But beyond this regime, when the interpore diffusion step C, dominates the mass uptake rate, the uptake rate slowed down for $PL_{N=N}$ as compared to that of $PL_{C=C}$ for similar thickness of the films (~ 530 ± 45 nm, saturation mass uptake time is ~3× slower for $PL_{N=N}$ compared to that of $PL_{C=C}$) (see Supplementary Fig. 4). These observations indicate that at lower mass loading, i.e., when methanol molecules are mostly near the surface, diffusivity rates are controlled by the pore windows which are similar in $PL_{C=C}$ and $PL_{N=N}$, i.e., $W_V$. While the above statement is true for an ideal MOF structure, surface barrier effect cannot be neglected. To evaluate this, we have carried out thin film thickness dependent methanol mass uptake measurements for both of the PL structures. The coinciding plots of fractional mass uptake versus normalized time indicate that surface barrier effect is not the controlling factor (Fig. 3b, c)[22]. This is because the uptake time follows a quadratic relation with the thickness of the film as shown in Eq. (2). We attribute this feature of the MOF thin film to the synthetic conditions used in this work (see synthesis methods). Hence, it is safe to assume that at lower mass loading vertically

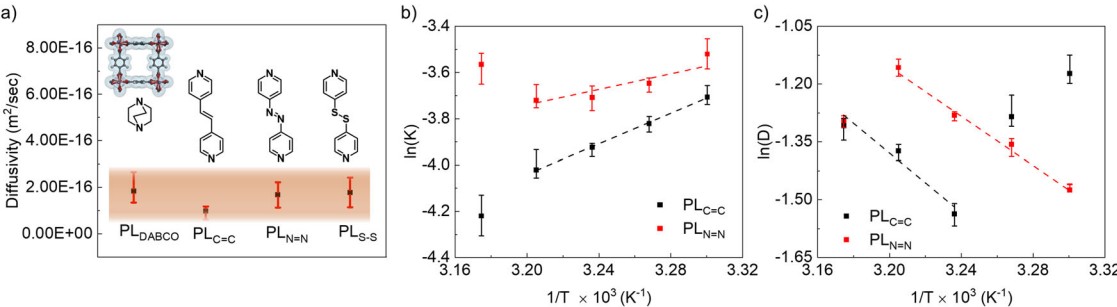

**Fig. 4 | Methanol diffusivity. a** Comparison of the diffusivities at 298 K for PL MOFs with oriented pores; the specific van der waals surface added pore is shown in the inset; for all the PLs with different pillars the accessible pores at the surface are similar; chemical structure of the different pillars are shown in the inset, **b** Arrhenius plot of diffusivity and **c** equilibrium constant. Error bars calculated using fitting parameters.

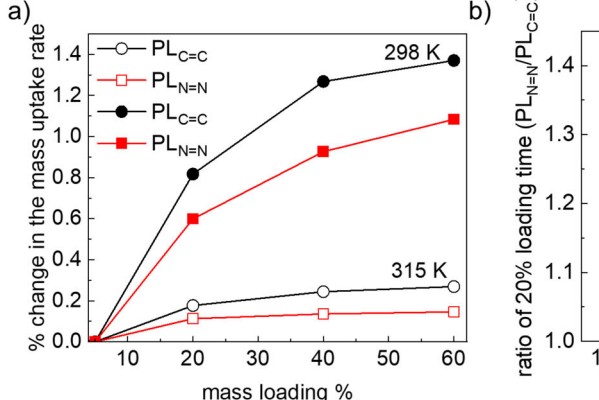

**Fig. 5 | Methanol uptake at variable temperature and concentration gradient. a** % changes in the mass uptake rates at different temperatures for the $PL_{C=C}$ and $PL_{N=N}$, **b** ratio of 20% loading time for $PL_{N=N}$ vs. $PL_{C=C}$ at different concentration gradient controlled by varying nitrogen flow. (See source data for details).

aligned $W_V$ is the key parameter.

$$m(t) = m_{eq}\left(1 - \exp\left(-\frac{t}{\tau}\right)\right) \qquad (2)$$

$$\tau = \frac{l^2}{3D} + \frac{l}{\alpha}$$

Where $m(t)$ denotes the mass uptake defined by an exponential decay function where $l$ denotes the thickness of the film, $D$ is the diffusivity, $\alpha$ is surface permeability, $m_{eq}$ is the equilibrium loading and $t$ is the time.

Note that at lower mass uptake regime, concentration gradient is maximum, and it is anticipated that methanol molecules will diffuse along the gradient through $W_V$. The surprising large difference in the uptake saturation time indicates that the larger windows $W_H$ do play an important role even though diffusion through these windows is orthogonal to the concentration gradient. Moreover, the $W_H$ sizes are similar for $PL_{C=C}$ and $PL_{N=N}$, hence it must be the different chemical functionalities that are controlling the diffusion rate. In the following discussion, we reveal that diffusion through $W_H$ is indeed rate limiting for the interpore diffusion and can be tuned by chemical design.

To ascertain that the dominating diffusion path for steps A and B involves $W_V$ only, we have carried out two sets of experiments: In experiment set 1, we have compared the methanol diffusivities of 4 isoreticular PL MOFs ($PL_{DABCO}$, $PL_{C=C}$, $PL_{N=N}$, and $PL_{S-S}$, see experimental section for details) having identical crystalline orientation, i.e., $W_V$ windows are vertically oriented. In these 4 PLs, $W_V$ dimensions and chemical functionalities are identical; however the $W_H$ windows are different. We have estimated similar diffusivity values for all the PLs at 298 K (Fig. 4a, see Supplementary Figs. 6 and 10). Hence no effect of

the $W_H$ can be observed. In experiment set 2, we have compared the activation energy ($E_A$) and enthalpy of adsorption ($\Delta H$) for the $PL_{C=C}$ and $PL_{N=N}$ for methanol (see Supplementary Figs. S11, S12) by measuring mass uptake at different temperatures. We found that the differences are very small ($\Delta H = -5.8$ ($\pm 0.5$) and 6.5 ($\pm 0.76$) kcal/mol and $\Delta E_A = 6.4$ ($\pm 0.57$) and 7.36 ($\pm 0.64$) kcal/mol for $PL_{N=N}$ and $PL_{C=C}$), Fig. 4b, c). The $E_A$ is estimated by the diffusivities at lower uptake regime, hence similar $E_A$ values confirm the hypothesis that steps A and B involve mostly $W_V$. Similar $\Delta H$ values indicate that the adsorbate-adsorbent interaction differences are small enough, to be identified by the present experimental setup. Density functional theory calculations indicate similar binding energies of methanol with $PL_{C=C}$ (13.87 kcal/mol) and $PL_{N=N}$ (13.35 kcal/mol), in accordance with the experimental observations.

After ruling out the surface barrier effect and confirming the role of $W_V$ at the lower mass loading, we focus on the differences observed at the higher mass loading. The distinct time differences in the saturation uptake can be attributed to the following structural features: (i) structural defect densities, (ii) cooperative effect between adsorbed molecules, (iii) lateral diffusion through $W_H$ pores with different functionalities. We rule out the defect densities, because in that case mass uptake rate will be affected also at the lower mass loading (steps A and B). To test the impact of cooperative effect, we have compared the percentage change in the rate of mass uptake (slope %) vs. fractional mass loading at two different temperatures (298 and 315 K, Fig. 5a). We observed that with increasing mass uptake, rate increases. It indicates that the methanol-methanol cooperative interaction at higher loading accelerates mass uptake. Furthermore, at lower temperature the change in the slope percentage is higher for both the PLs. This is probably due to the stronger

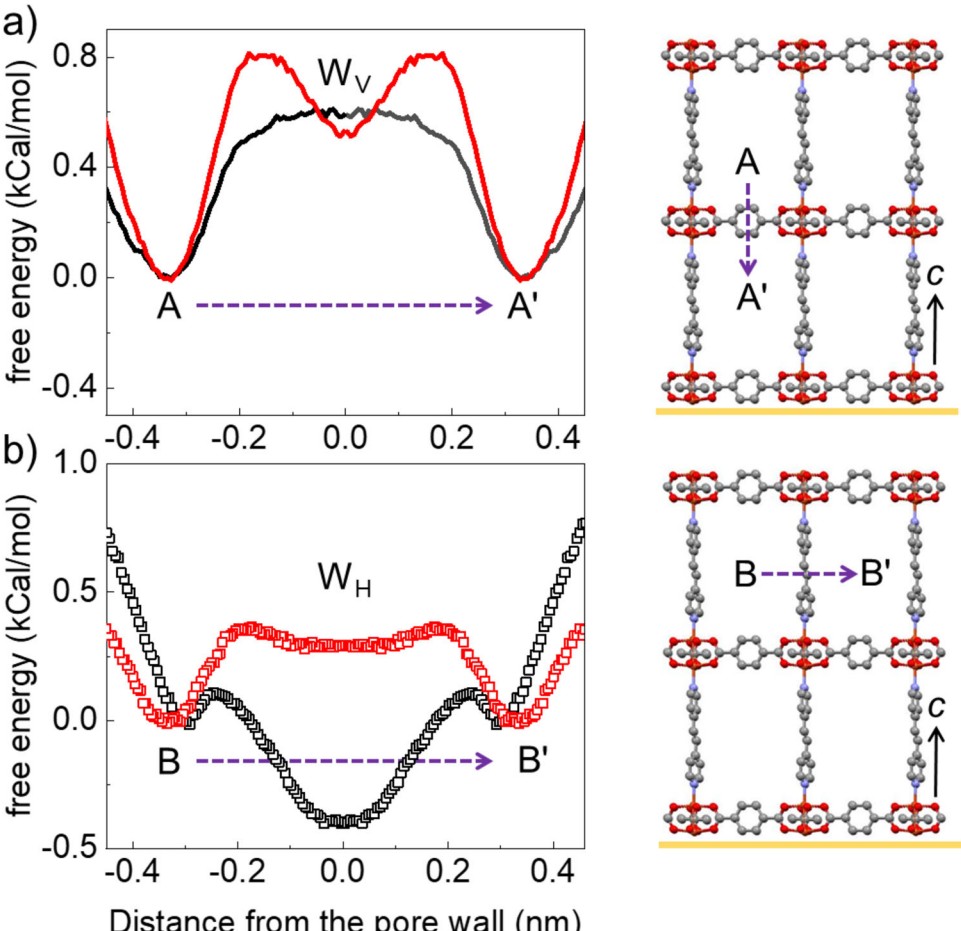

**Fig. 6 | Free energy profiles for interpore diffusion.** Free energy profiles of a methanol molecule during transition from one pore to the other through (**a**) $W_V$ and (**b**) $W_H$; red = $PL_{N=N}$ and black = $PL_{C=C}$. (Right) the net transition paths are shown in dotted lines for the $PL_{C=C}$ structure; molecular geometries corresponding to the energy minimum (A or A' and B or B' positions) are shown in Supplementary Figs 14–15.

methanol-methanol interaction at the lower temperature, indicating presence of similar cooperativity in both the PLs. Hence, methanol cooperative interaction is not the rate limiting step at higher mass loading.

In light of the dependence of interpore diffusivities on the $W_H$ windows, which are stationed orthogonal to the concentration gradient, we postulate that the effective diffusion is governed not only by the concentration gradient but also by the pore window size. At the lower uptake regime, during steps A and B, concentration gradient is highest and hence, it dominates the mass uptake rate. At higher mass loading (when the interpore diffusion dominates) concentration gradient continuously decreases, and the pore window size becomes the rate limiting factor. In the present case 3× larger size of $W_H$, as compared to $W_V$, dictates the diffusion path during interpore diffusion at lower concentration gradient. This is verified by concentration gradient dependent methanol mass uptake measurements (see Supplementary Fig. 13). As the concentration gradient reduces, differences in the uptake rates become more prominent even at the lower mass loading (<20%) (Fig. 5b). This phenomena can be generalized to any 3D porous structure, in which more than one type of pore window exists. Note that the net diffusion still follows the Ficks law, only the microscopic interpore diffusion path varies. Evidently, as the chemical functionalities of the $W_H$ are changed, uptake rates change sharply. Comparing the mass uptake time of methanol and 1-butanol, for the different PLs with different pillar functionalities indicate that (size-based) selectivities are higher at saturation, compared to the lower mass loading region (Supplementary Table 1). Using this approach, the

permeation and selectivity of the chemical mixtures can be regulated rationally, in a preconceived manner.

To get an insight into the energy barriers along the $W_V$ and $W_H$ pores for methanol, we performed force field based molecular dynamics simulations (see computational section). The comparative free energy profiles are illustrated in Fig. 6. It is evident from these simulations that during the diffusion along the $W_V$ pores (from A to A'), the free energy changes are similar for $PL_{C=C}$ and $PL_{N=N}$. On the contrary, it is energetically uphill to traverse along the $W_H$ pores (from B to B') for $PL_{N=N}$ but energetically downhill for $PL_{C=C}$. This horizontal movement allows a higher mass uptake rate in $PL_{C=C}$, that is observed in experiments. A preferential interaction between methanol and the pillar functionalities is clearly visible at the two energy minimum (see Figs. S14–S16) at 0.0 nm (at the window) and 0.3 nm (on the edges of the PL-cage). The potential energy difference calculated between the two positions ($PL_{C=C}$ (0.3–0.0 nm) = 0.5 kcal/mol, $PL_{N=N}$ (0.3–0.0 nm) = −0.24 kcal/mol) also shows preferential movement of methanol molecules across the $PL_{C=C}$ MOFs, ascertaining the hypothesis of chemically controlled interpore diffusion.

## Discussion

Complexity in microscopic mechanism for interpore diffusion, which is the rate determining step during the permeation through a porous membrane or during a catalysis process, is challenging to resolve. This lack in clarity is due to the fact that the simplistic model of concentration gradient-dependent diffusion does not strictly apply. By careful analyses of the mass uptake rates in the oriented nanochannels

of the pillared-layer MOFs, we could reveal the diffusion path and rate limiting parameters. We have demonstrated that surface barrier effect is absent in this MOF thin films, as the preparation method is carefully optimized. Different types of pillared-layer MOFs were grown in a layer-by-layer fashion as oriented thin films, and this allowed correlating the mass uptake rates with chemical functionalities and pore window orientation. The experimental observations indicated that in spite of the presence of concentration gradient, diffusivity is controlled by the large pores aligned along the direction orthogonal to the gradient. The diffusion directionality is dependent on both, the chemical gradient and the pore window size, and these factors dominate at different loading %. The changes in chemical functionality in these pore windows, realized by changing the pillar functionalities of the MOFs, drastically modulate the uptake time, resulting in a chemical control of the overall molecular diffusion. In the present case, we have found that ethylene (-C = C-) functionality, in comparison to -N = N- and -S−S-, helps to accelerate the mass uptake rate. The applicability of this diffusion mechanism can be extended to other adsorbate molecules and porous solids, in which the pores are 3D and surface barrier is negligible. However, nature of adsorbate-adsorbent interactions can vary in a nonlinear fashion and hence diffusivity rates will change accordingly[31]. Note that flexibility (local and global) of the MOF structures can also influence diffusivity, and this aspect is not addressed in the current hypothesis. The insight of the diffusion path and the chemical route to modulate diffusion can be applied further to designing of nanoporous membranes for chemical separation, e.g., aliphatic and aromatic hydrocarbons, pollutant gases and volatile organic compounds[55]. Also the chemical reactions carried out in the confined spaces of porous catalysts can be tuned using the findings presented here.

## Methods
### Synthesis of 4,4'-azopyridine
4,4'-Azopyridine was synthesized following a reported method[56].

### Synthesis of pillared-layer MOF thin films on QCM substrate
5 MHz (gold coated) QCM-sensors were dipped in an ethanolic solution (20 mM) of 11-mercapto-1-undecanol (MUD) for 24 h to obtain −OH functionalized surface. These substrates were then thoroughly washed with absolute ethanol (99.99%), dried and used for thin films synthesis. SiO₂/Si substrates were cleaned by isopropanol and then by UV-ozone cleaner, to remove organic impurities and to create free −OH groups on the surface. The MOF thin films were prepared on those functionalized substrate via a well-known layer-by-layer (lbl) liquid-phase epitaxial (LPE) method[48]. The method consists of four steps to complete a cycle at 60 °C as: (i) dipped in 1 mM copper acetate ethanol solution for 10 min, (ii) drained the metal solution and washed with fresh ethanol, (iii) dipped in 0.2 mM linker solution (mixture of two linkers) in ethanol for 20 min and (iv) drained the linker solution and washed with fresh ethanol. MOF thin films with varying thickness were prepared by varying the number of cycles. 1,4-Benzene dicarboxylic acid is the primary linker used with different pillar linkers (4,4'-azopyridine, 1,2-di(4-pyridyl)ethylene, 4,4'-dithiodipyridine and 1,4-diazabicyclo[2.2.2]octane) for MOF films.

### Characterizations
Powder X-ray diffractometer (XRD) patterns of thin films were recorded on a Rigaku XDS 2000 diffractometer using nickel-filtered Cu Kα radiation (λ = 1.5418 Å) ranging from 5 to 20° at room temperature (voltage 40 kV, current 200 mA). Out-of plane PXRD was recorded in 2θ/θ (step size 0.01, scan rate 0.2°/s), in-plane in 2θ/φ geometry with grazing incident angle (ω) at 0.3° and step size of 0.01 with scan rate 0.1°/s.

Surface morphology of samples were characterized using field emission scanning electron microscopy (FESEM), JEOL JSM-7200F

instrument with a cold emission gun operating at 30 kV. Energy-Dispersive X-ray spectroscopy (EDS) elemental analysis and mapping were also done on the FESEM.

The vibrational Raman spectra were recorded by using a Renishaw inVia Raman microscope (532 nm excitation).

The adsorption profiles were measured using a quartz crystal microbalance (QCM) from open QCM, Italy. Thickness for all the thin films was calculated using J.A. Wollam ellipsometer (alpha-SE). The data was fitted using a B-Spline model including surface roughness.

**QCM experiments.** The MOF samples were activated preceding the measurements by heating the QCM sensors at 65 °C for 12 h under vacuum (10⁻⁴ bar). Mass uptake experiments were carried out using a constant flow rate (50 sccm) of dry N₂, passing through saturated solvent vapors (methanol, 1-butanol).

**Analyses of uptake kinetics.** Mass-frequency relationship for the QCM measurements is given by Sauerbrey equation;[28]

$$\Delta m = - c \frac{\Delta f}{n} \qquad (1)$$

Where $n$ denotes the overtone order ($n$ = 3, 5, and 7) and c is the mass sensitivity constant. For a 5 MHz crystal, c has value of 17.7 ng/cm².

Diffusivity, $D$, can be obtained by fitting the mass uptake vs. the square root of adsorption time using this equation:[28]

$$\frac{M_t(t)}{M_\infty} \approx \frac{8}{\sqrt{\pi}} \frac{\sqrt{Dt}}{L^2} \qquad (3)$$

**Computational details.** All the periodic DFT calculations were performed using PBE functional along with empirical D3 correction as implemented in CP2K software package that employs Gaussian plane waves. Double zeta quality basis sets were employed for all the atoms (DZVP-GTH-qn for all non-metallic atoms and DZVP-MOLOPT-SR-GTH for the Cu centers) along with GTH-PBE pseudopotentials. For the PL$_{C=C}$ and PL$_{N=N}$ structures geometry and cell parameters were optimized simultaneously. In order to run force field-based molecular dynamics simulations RESP fitted partial charges were computed with REPEAT method using Bloechl charges as initial guess. Coordinates of the structures are provided in the Supplementary Information. The binding energy of methanol with the PL MOFs (E$_{binding}$) is computed as follows:

E$_{binding}$ = <E$_{system}$> - E$_{MeOH}$ − E$_{MOF}$, where for the average system energy single point DFT calculations are performed on multiple MD snapshots. Atomic coordinates used for the binding energy calculation is provided as an additional file.

**Molecular dynamics.** The MOF structures were constructed by multiplying the ab-initio optimized 1 × 1 × 1 unit cell, to create 3 × 3 × 3 cages that are periodic along *ab* and terminated along *c* with a vacuum. UFF LJ[57] parameters and ab-initio computed charges (Supplementary Table 2) are used to simulate the non-bonding interactions of the MOF system, while the MOF is considered frozen. For the methanol molecule, CHARMM parameters are calculated using CHARMM-GUI[58]. All simulations are performed in the open source program GROMACS[59].

100 methanol molecules are added to the MOF system and the system is equilibrated. NVT ensemble simulations are performed, with a temperature of 300 K maintained using the V-rescale method[60]. In case of 100 methanol molecules, a longer (1 μs) simulation is performed to generate the density distribution shown in Supplementary Fig. 17. For the free energy profiles, umbrella sampling simulations[61] are performed that bias the perpendicular distance (along *a* or *c* axis) between the pore (W$_H$ or W$_V$) and the methanol molecule. The W$_V$ and W$_H$ profiles contain 11 (0.0−1.0 nm) and 7 (0.0−0.6 nm) sampling

windows each (of 100 ns each) that are 0.1 nm apart, and apply a bias of 100–300 kJ/mol. A smaller bias (50 kJ/mol) is also added to the parallel direction to restrict greater movement of the methanol molecule.

## Data availability

All data generated or analyzed during this study are included in the published article, supplementary dataset 1 and source data file. Source data are provided with this paper.

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

## Acknowledgements
We acknowledge intramural funds at TIFR Hyderabad from the Department of Atomic Energy (DAE), India, under project identification number RTI 4007. R.H. and J.M. also acknowledge Infosys-TIFR leading edge grant (cycle 2) for financial support.

## Author contributions
T.M. performed the experiments; P.M. analyzed the data; structural simulations, calculations, and analyses were performed by S.B., S.G., and J.M., R.H. conceived the idea. T.M., P.M., and R.H. contributed to the discussion and writing of the manuscript with inputs from all the coauthors.

## Competing interests
The authors declare no competing interests.
