## [Peer Review File · Nature Communications]

REVIEWERS' COMMENTS:

Reviewer #1 (Remarks to the Author):

Review on "Chemically routed interpore molecular diffusion in nanoporous thin films" by T. Maity et al. for Nature Communications

By exploring the elementary steps of diffusion in nanoporous materials the authors deal with a hot topic. They, moreover, report about interesting results. However, their discussion and interpretation in the given paper contains – at least in my understanding – some deficiencies, which exclude their publication in the present form.

First of all, I came across a number of obvious carelessness errors. They include, for example, the legend of Figure 2 with a confusion of subfigures (c) and (d), some of the literature references (3, 29) that were incorrect or incomplete, and Figure S8 and some of the subsequent figures where the abscissa notation (physical quantities vs. their units) is – at least – unusual.

My main concern, with respect to the clarity of presentation, refers to the use of the terms "windows" and "pores". In my understanding, following the scheme shown in Figure 1, there are two types of windows, but only one type of pores (namely the one shown in Fig. 1), with the two types of windows (rather than "pores", as indicated in the legend and, as well, repeatedly in the text) called WV and WH.

My main criticism with reference to the logics concerns the reasoning as provided in the second paragraph of the Introduction. Here (in lines 47, 48) it is said that step C (i.e. interpore diffusion) is usually the rate-limiting factor in uptake/permeation (a statement which is, obviously, implied throughout the paper) and that (lines 51, 52) problems in rationalizing diffusion in nanoporous materials arise as soon as nonlinearity effects (i.e. a concentration-dependent diffusivity) must be considered. I agree with this statement in so far that the mathematics in dealing with concentration-dependent diffusivities becomes more complicated, but the "physics" behind the phenomenon remains the same.

In the given paper, the authors try to confirm the need for a new model by reporting a fascinating result of their measurement, namely an enhancement of the rate of uptake by an enhancement of the diameters of the windows giving rise to mass transfer in the direction perpendicular to the flow direction through different pillared layer structures, while the diameters of the windows in flow direction have remained unchanged.

Though, at a first glance, this finding may indeed be found to be puzzling, it can be referred to existing knowledge about mass transfer in the considered systems. First, it is very important to have in mind that there is no firm foundation allowing to assume that, as a rule, the influence of surface barriers (see, e.g., D.M. Ruthven: "Diffusion in type A zeolites: New insights from old data", *Micropor. Mesopor. Mat.* 162 (2012) 69–79., D.M. Ruthven et al.: "Sorption kinetics: measurement of surface resistance", *Adsorption* 27 (2021) 787–799). on uptake/permeation is negligibly small. Surface barriers become notably relevant for thin-layered structures (see, e.g., ref. [22] of the present paper) since the relative contribution of surface barriers on overall uptake/permeation increases with decreasing layer thickness. The huge difference between an "apparent" diffusivity of about 6×10^{-15} as obtained in the present studies for methanol in ZIF-8 (Fig. 2b) and the data "microscopically" determined ($\sim 10^{-11}$, C. Chmelik: "Characteristic features of molecular transport in MOF ZIF-8 as revealed by IR microimaging", *Micropor. Mesopor. Mat.* 216 (2015) 138–145) and confirmed by transition state theory (C. Chmelik et al.: "The predictive power of classical transition state theory revealed in diffusion studies with MOF ZIF-8", *Micropor. Mesopor. Mat.* 225 (2016) 128–132) suggest, that also in the given case mass transfer is mainly controlled by surface permeation.

Surface barriers on nanoporous materials have been found to be formed by either a homogeneous layer of dramatically reduced permeability (see, e.g.: J. Cousin Saint Remi et al.: "The role of crystal diversity in understanding mass transfer in nanoporous materials", *Nat. Mater.* 15 (2015) 401–406.) or by an essentially impermeable layer with statistically distributed small holes (F. Hibbe et al.: "Monitoring Molecular Mass Transfer in Cation-Free Nanoporous Host-Crystals of Type AlPO-LTA," *J. Am. Chem. Soc.* 134 (2012) 7725–7732.).

It is precisely this second type of surface barrier that would explain the experimentally observed behavior: The faster the molecules can diffuse parallel to the surface (i.e., perpendicular (!) to the direction of flow), the more likely they should be able to reach a hole in the otherwise impermeable surface cover. A related example, where increased diffusion in a particular direction

helps to increase diffusion in the perpendicular direction as well, may be found in D. Kondrashova et al.: "Scale-dependent diffusion anisotropy in nanoporous silicon", Sci. Rep. 7 (2017) 40207. While I would highly recommend a re-examination of the results obtained in light of these comments with a corresponding reformulation of the discussion and conclusions derived, I am very skeptical about publishing it in its current form, especially given the rank of the journal to which the paper was submitted.

Reviewer #2 (Remarks to the Author):

The manuscript "Chemically routed interpore molecular diffusion in nanoporous thin films" seeks to understand the role of chemical functionality in pillar molecules on the interpore diffusion process. It was shown that at high mass loading, the interpore diffusion is controlled by lateral diffusion, which is greatly affected by the pillar. The pillar molecule Py-C=C-Py shows a higher uptake rate than Py-N-N-Py and Py-S-S-Py pillars, which is then verified by the molecular dynamic simulations. It was also shown that the size-based selectivity is higher at the high mass loading (saturation) region. This work offers an understanding of the diffusion process happening in the 3D porous structure and provides possible design principles for the diffusion happening in porous structures. However, the authors failed to give any possible explanation for the selectivity coming from the pillar's chemical properties (electronic effect? hydrogen bonding?). The information present is of value to the scientific community, but it is unclear whether the results rise to the level of Nature Communications.

Specific Comments:

1. (Page 9, lines 169-175) -It would be better to denote for Py-S-S-Py just like C-C and N-N.
2. (Page 8, Fig. 2.D) - Authors should explain the large deviation for the data points in the upper right corner
3. (Page 9, lines 214-216) - Is the case contrary to Fickian diffusion? There is no evidence showing that diffusion is against the concentration gradient. Additionally, is it possible to have the concentration gradient in the a,b-plane caused by the edge effect (in other words, the methanol diffuses from the side wall)?

Reviewer #3 (Remarks to the Author):

This paper concentrated on the interpore molecular diffusion in nanoporous. It is quite interesting topic, but the results actually did not surprise me at all. Such as the interpore diffusion is the rate-limiting step, which is not a new viewpoint. Maybe the novelty is that the authors design the experiments to prove this point. Another, the title emphasized the nanoporous, instead of MOF nanoporous. It is not strict. To my knowledge, MOF is quite special, comparing with other porous materials. Other questions were shown as below:

- (1) Can you provide more SEM or other microscopic graphs to show the oriented MOFs? Such as those works by J. Caro.
- (2) How to understand the negative energy barriers in Figure 3(b)? Attractive interactions for the inter-pore diffusion? It is confusing!
- (3) Flexibility, linker swing or structure transformation, is considered as one of the most attractive characters of MOFs, comparing to other porous materials. It is believed that the flexibility will influence the diffusivity of guests and then the mixture separation. The authors should discuss this point here.
- (4) Small grammar errors, such as "Similar ΔH values indicates that.....".
- (5) Because the MOF is considered frozen in MD simulations, I doubt the free energy values in this work. Especially the free energy difference is quite small here.

Hence, I do not think this work in current form can be acceptable in Nature Communication.

Point-by-point response

Reviewer #1 (Remarks to the Author):

Review on "Chemically routed interpore molecular diffusion in nanoporous thin films" by T. Maity et al. for Nature Communications
By exploring the elementary steps of diffusion in nanoporous materials the authors deal with a hot topic. They, moreover, report about interesting results. However, their discussion and interpretation in the given paper contains – at least in my understanding – some deficiencies, which exclude their publication in the present form. First of all, I came across a number of obvious carelessness errors. They include, for example, the legend of Figure 2 with a confusion of subfigures (c) and (d), some of the literature references (3, 29) that were incorrect or incomplete, and Figure S8 and some of the subsequent figures where the abscissa notation (physical quantities vs. their units) is - at least - unusual.

Response: Thank you for the comments and suggestions. As per suggestions, we have rechecked the interpretations, and now we have revalidated the proposed hypothesis. Please see our responses which are appended below.

These errors are now corrected in the revised version and highlighted in yellow.

My main concern, with respect to the clarity of presentation, refers to the use of the terms “windows” and “pores”. In my understanding, following the scheme shown in Figure 1, there are two types of windows, but only one type of pores (namely the one shown in Fig. 1), with the two types of windows (rather than “pores”, as indicated in the legend and, as well, repeatedly in the text) called WV and WH.

Response: This particular aspect has been rectified. We have corrected and highlighted the changes in yellow.

My main criticism with reference to the logics concerns the reasoning as provided in the second paragraph of the Introduction. Here (in lines 47, 48) it is said that step C (i.e. interpore diffusion) is usually the rate-limiting factor in uptake/permeation (a statement which is, obviously, implied throughout the paper) and that (lines 51, 52) problems in rationalizing diffusion in nanoporous materials arise as soon as nonlinearity effects (i.e. a concentration-dependent diffusivity) must be considered. I agree with this statement in so far that the mathematics in dealing with concentration-dependent diffusivities becomes more complicated, but the “physics” behind the phenomenon remains the same.

Response: We agree with the comments. To state our point of view clearly, we have omitted the sentence “Hence, a rationale to experimentally map interpore diffusion becomes more difficult. The validity of the Fickian model becomes more problematic”.

In the given paper, the authors try to confirm the need for a new model by reporting a fascinating result of their measurement, namely an enhancement of the rate of uptake by an enhancement of the diameters of the windows giving rise to mass transfer in the direction perpendicular to the flow direction through different pillared layer structures, while the

diameters of the windows in flow direction have remained unchanged. Though, at a first glance, this finding may indeed be found to be puzzling, it can be referred to existing knowledge about mass transfer in the considered systems. First, it is very important to have in mind that there is no firm foundation allowing to assume that, as a rule, the influence of surface barriers (see, e.g., D.M. Ruthven: “Diffusion in type A zeolites: New insights from old data”, *Micropor. Mesopor. Mat.* 162 (2012) 69–79. , D.M. Ruthven et al.: “Sorption kinetics: measurement of surface resistance”, *Adsorption* 27 (2021) 787–799). on uptake/permeation is negligibly small.

Response: We agree that surface barrier affect might be present, and not necessarily negligible. This is also stated in the revised version of the manuscript (Introduction, 2nd paragraph)

Surface barriers become notably relevant for thin-layered structures (see, e.g., ref. [22] of the present paper) since the relative contribution of surface barriers on overall uptake/permeation increases with decreasing layer thickness. The huge difference between an “apparent” diffusivity of about 6×10^{-15} as obtained in the present studies for methanol in ZIF-8 (Fig. 2b) and the data “microscopically” determined ($\sim 10^{-11}$, C. Chmelik: “Characteristic features of molecular transport in MOF ZIF-8 as revealed by IR microimaging”, *Micropor. Mesopor. Mat.* 216 (2015) 138–145) and confirmed by transition state theory (C. Chmelik et al.: “The predictive power of classical transition state theory revealed in diffusion studies with MOF ZIF-8”, *Micropor. Mesopor. Mat.* 225 (2016) 128–132) suggest, that also in the given case mass transfer is mainly controlled by surface permeation.

Response: The ZIF-8 and the pillared-layer MOFs cannot have similar surface barriers. This is because of their different chemical components and different synthesis conditions. And hence a direct reference to those differences in the diffusivity values will not help understanding the present case. To avoid any confusion, we have omitted the ZIF-8 experimental data from the manuscript and focus only on the pillared layer MOFs. In the following we also prove that in the present study, surface barrier effect is actually negligible.

Surface barriers on nanoporous materials have been found to be formed by either a homogeneous layer of dramatically reduced permeability (see, e.g.: J. Cousin Saint Remi et al.: “The role of crystal diversity in understanding mass transfer in nanoporous materials”, *Nat. Mater.* 15 (2015) 401–406.) or by an essentially impermeable layer with statistically distributed small holes (F. Hibbe et al.: “Monitoring Molecular Mass Transfer in Cation-Free Nanoporous Host-Crystals of Type AlPO-LTA,” *J. Am. Chem. Soc.* 134 (2012) 7725–7732.).

It is precisely this second type of surface barrier that would explain the experimentally observed behavior: The faster the molecules can diffuse parallel to the surface (i.e., perpendicular (!) to the direction of flow), the more likely they should be able to reach a hole in the otherwise impermeable surface cover. A related example, where increased diffusion in a particular direction helps to increase diffusion in the perpendicular direction as well, may be found in D. Kondrashova et al.: “Scale-dependent diffusion anisotropy in nanoporous silicon”, *Sci. Rep.* 7 (2017) 40207.

Response: This proposed alternative explanation is valid if we can find an evidence of only surface barrier dominated diffusion phenomenon. The following experimental evidences indicate that surface barrier is not the dominant aspect:

Experiment 1: It is known in the MOF thin film literature that by optimizing a synthesis condition and for thicker MOF thin films, surface barriers can become negligible. Please see: <https://pubs.acs.org/doi/full/10.1021/cm071632o>. In the present case, the synthesis conditions, a layer-by-layer deposition of the thin film, are optimized to achieve defect reduced structure (<https://www.nature.com/articles/ncomms5562>). To find out that whether the surface barriers are present, we have investigated the mass uptake rates in the thin films of different thickness. In absence of surface barrier effect, uptake rate should be quadratically dependent on the film thickness (<https://www.nature.com/articles/ncomms5562>). We found that the plots of fractional mass uptake vs $\sqrt{\text{time}/\text{thickness}}$, for different thicknesses of the pillared-layer MOFs coincide, confirming that the surface barriers are negligible. This experimental data is presented in Figure 2b and 2c in the manuscript.

Experiment 2: We have carried out a mass uptake experiment, in which we have varied the concentration gradient along the thickness of the film. We have observed that at lower concentration gradient (at lower loading percentage), the difference in the uptake rates between the two different pillared MOF thin films become significant. At ~60% loading the mass uptake time differs by 2-times. This observation is highly unlikely if the mass loading is only dominated by the surface barrier effect, as postulated by the reviewer 1, since the defect densities in the two systems are not supposed to be drastically different given similar synthesis procedures. The strong concentration gradient dependency strengthens our hypothesis that at high loading percentage diffusion is dominated by the vertically aligned small pore windows, while during the pore-to-pore diffusion larger chemically functionalized pore windows control the rate. Related figure is appended in the supporting information, Figure S13.

Based on these experimental evidences we ascertain that the pore to pore diffusion, along the horizontal direction of the gradient dictates the saturation uptake. This is reflected by changing of the chemical functionalities, as presented in this paper.

Reviewer #2 (Remarks to the Author):

The manuscript “Chemically routed inter-pore molecular diffusion in nanoporous thin films” seeks to understand the role of chemical functionality in pillar molecules on the inter-pore diffusion process. It was shown that at high mass loading, the inter-pore diffusion is controlled by lateral diffusion, which is greatly affected by the pillar. The pillar molecule Py-C=C-Py shows a higher uptake rate than Py-N-N-Py and Py-S-S-Py pillars, which is then verified by the molecular dynamic simulations. It was also shown that the size-based selectivity is higher at the high mass loading (saturation) region. This work offers an understanding of the diffusion process happening in the 3D porous structure and provides possible design principles for the diffusion happening in porous structures.

However, the authors failed to give any possible explanation for the selectivity coming from the pillar’s chemical properties (electronic effect? hydrogen bonding?). The information present is of value to the scientific community, but it is unclear whether the results rise to the level of Nature Communications.

Response: The novelty of the work is realization of diffusion controlling factor in heterogeneous pore system and chemical tuning of those factors. Hence, we did not highlight the “specific interactions” that determines the diffusion selectivity.

In fact, our experimental observations suggest that the differences in the binding enthalpy and activation energy in the pillared-layer MOFs are negligible, and density functional theory calculations support this fact. Please see pages number 8-9. Hence, the size of the pore windows is the key parameters here.

Further, in the classical MD free energy profile shown in figure 5, a clear difference is seen in translation of methanol cross the ethyl and azo pillared MOFs. While chemical interactions like hydrogen bonding with the OH groups of methanol and partial charge interactions with the ligand groups are expected to play a part, the calculation cannot determine a precise contributor to the interaction, primarily due to the much larger size of the pore and the dynamic nature of the MD simulations that take into account various orientations in the output energy.

To provide a clear view of the possible interactions at the pore, we have provided snapshots of the methanol molecules inside the pores, in supporting information figures S14 and S15.

Specific Comments:

1. (Page 9, lines 169-175) -It would be better to denote for Py-S-S-Py just like C-C and N-N.

Response: According to the suggestion, it has been updated in the manuscript.

2. (Page 8, Fig. 2.D) - Authors should explain the large deviation for the data points in the upper right corner

Response: We would like to thank the reviewer for raising this point. In the Arrhenius plots of Figure 3b and c five data points were recorded by measuring the mass uptake at different temperatures and out of which the best four points following a linear trend were used to calculate the ΔH and ΔE_A respectively. The point showing large deviation was not considered for the fitting. Since the diffusivity and mass uptake values used to calculate ΔH and ΔE_A are not absolute and the diffusivity values are calculated from low uptake regime only therefore sometimes deviation in data point can be expected. The values obtained shows a very small difference for both $PL_{C=C}$ and $PL_{N=N}$ confirming the similar adsorbate-adsorbent interactions in the low uptake region.

3. Fig 3. (Page 9, lines 214-216) – Is the case contrary to Fickian diffusion? There is no evidence showing that diffusion is against the concentration gradient. Additionally, is it possible to have the concentration gradient in the a,b-plane caused by the edge effect (in other words, the methanol diffuses from the side wall)?

Response: No, the observed results are in accordance to the Fickian diffusion. This is also stated in the manuscript. Our experimental observations explain that the path of the molecular diffusion at the higher loading is determined by the larger sized pores, which are orthogonal to the gradient. However, the net mass diffusion is following Fick's law.

It is not possible to have concentration gradient in the *ab* plane in our current setup. This is because in our experimental setup the saturated vapours of methanol are flowing vertically across the film thickness and ruling out the effect of diffusion through the side walls.

Reviewer #3 (Remarks to the Author):

This paper concentrated on the inter-pore molecular diffusion in nanoporous. It is quite interesting topic, but the results actually did not surprise me at all. Such as the inter-pore diffusion is the rate-limiting step, which is not a new viewpoint. Maybe the novelty is that the authors design the experiments to prove this point. Another, the title emphasized the nanoporous, instead of MOF nanoporous. It is not strict. To my knowledge, MOF is quite special, comparing with other porous materials. Other questions were shown as below:

Response: The novelty of this work is not that the pore to pore diffusion is rate controlling step. We have shown that the rate controlling diffusion is orthogonal to the chemical bias, due to the presence of heterogeneous (large and small pores) pore system. The novelty of the work is realization of diffusion controlling factor in heterogeneous pore system and chemical tuning of those factors.

According to the suggestion we have modified the manuscript title.

(1) Can you provide more SEM or other microscopic graphs to show the oriented MOFs? Such as those works by J. Caro.

Response: According to the suggestion, we have modified the SEM picture, which shows the cross section of the thin film (Figure 1a).

SEM images cannot confirm the orientation of the MOF. To confirm that, we have carried out in- and out-plane x-ray diffraction measurements (see Figure 1b).

(2) How to understand the negative energy barriers in Figure 3(b)? Attractive interactions for the inter-pore diffusion? It is confusing!

Response: Figure 3(a) and 3(b) (in the revised version these are 5a-b) plot the free energy landscape of a methanol molecule across W_V and W_H . The free energy values themselves are not absolute and are relative to each other. To visualize and compare the results between different systems, the free energy value for a position that is equivalent between the two systems (at ~ 0.3 nm from the pore) is set equal to 0, to compare the transport behaviour in a better way.

For the movement of a methanol molecule between the minima points B and B', it is energetically uphill to traverse along the W_H pores for $PL_{N=N}$ but energetically downhill for $PL_{C=C}$. This implies that due to the shift in the $PL_{C=C}$ local minima to the W_H pore, the methanol molecule can traverse easily between pores, while it is expected to get stuck at the ~ 0.3 nm position in case of $PL_{N=N}$.

(3) Flexibility, linker swing or structure transformation, is considered as one of the most attractive characters of MOFs, comparing to other porous materials. It is believed that the flexibility will influence the diffusivity of guests and then the mixture separation. The authors should discuss this point here.

Response: This is a very important aspect of MOFs. In the present case we have not considered this parameter, and rather focussed on the pore window size and orientation. In future we would

like to consider the flexibility aspect, and look into the diffusion features more carefully. Inclusion of these parameters is beyond the scope of the current work.

As per suggestion, we have noted the flexibility aspect in the revised manuscript and highlighted in yellow.

(4) Small grammar errors, such as “Similar ΔH values indicates that.....”.

Response: This has been corrected in the revised version.

(5) Because the MOF is considered frozen in MD simulations, I doubt the free energy values in this work. Especially the free energy difference is quite small here.

Response: Flexibility and movement of the side groups can affect the movement of interacting molecules across a pore. This is highly important in the movement of molecular sizes comparable to the pore size where the rotation of side groups can limit and control the movement of molecules. This effect, however, becomes less important as the molecular size becomes much smaller than the pore size. In the current work, there is no steric barrier caused by the side groups, as can be seen by the relative sizes of methanol compared to the W_H and W_V pores in Figures S14 and S15. The ab-initio optimized $PL_{C=C}$ and $PL_{N=N}$ structures correspond to local minima, which are taken as representative frozen structures for the MD simulation. Also, any minor effects pertaining to rotation of side groups and resultant steric interactions in these systems should be very similar for $PL_{C=C}$ and $PL_{N=N}$ MOFs and would not affect the energetic comparison made here.

REVIEWERS' COMMENTS:

Reviewer #1 (Remarks to the Author):

Review on "Chemically routed inter-pore molecular diffusion in metal-organic framework thin films" by T. Maity et al. for Nature Communications

I appreciate the care spent by the authors on revising their paper and in responding to the reviewers' comments. I was, in particular, pleased to see that, by considering the dependence of uptake time on the material thickness (Figs. 2b,c), the authors were able to exclude any significant influence of surface barriers on the rate of molecular uptake. All the more spectacular seems to me the finding that an increase of the rate of mass transfer perpendicular to the concentration direction and thus perpendicular to the direction of the molecular flux can lead to an increase of this very flux. In a possibly over-simplified picture, I take it as an indication that the probability of molecular propagation (i.e. of window passages) in flux direction is not uniform and varies, e.g., with the concentration in the cages into which the molecules are going to jump. In such a scenario, an enhancement of diffusion perpendicular to the flux direction would indeed be understood to, possibly, as well enhance the jump rate in flux direction, i.e. to give rise to flux enhancement. This would, moreover, nicely correlate with the diffusivity enhancement with increasing loading (Fig. 4).

I guess, after the revision the present paper is well suited for publication in Nature Communications. Leaving it to the authors to possibly refer to also the present considerations, I warmly recommend acceptance.

Reviewer #3 (Remarks to the Author):

Although this manuscript has been improved according to the previous reviewing, still I do not think it can reach the level of Nature Communications.

The novelty of this work, as you say, is the realization of diffusion controlling factor in heterogeneous pore system and chemical tuning of those factors. In my opinion, it is a little exaggeration. I do not think the current research results can support this novelty. Honestly, this is a nice try and some interesting results are presented. But Nature Communications is a top journal and only publishes high-quality research. Current work should be submitted to other specialized journal.

The absolute values of free energy are meaningless, please use the relative values in Figure 5 to avoid the confusion.

Regarding to your opinion about the effect of MOF flexibility, I can not agree it completely. In Figures S14 and S15, only one guest molecule is presented, as is not the real case. It means that you did not consider the concentration of guests inside MOF pores. It is not rigorous to say "there is no steric barrier caused by the side groups....". Maybe this effect of MOF flexibility will not alter current conclusion, but this kind of effect is indeed existed.

In conclusion, Nature Communications is an excellent journal and this manuscript is at this point not of the necessary caliber.

Point-by-point response:

Reviewer 1:

Review on "Chemically routed inter-pore molecular diffusion in metal-organic framework thin films" by T. Maity et al. for Nature Communications. I appreciate the care spent by the authors on revising their paper and in responding to the reviewers' comments. I was, in particular, pleased to see that, by considering the dependence of uptake time on the material thickness (Figs. 2b,c), the authors were able to exclude any significant influence of surface barriers on the rate of molecular uptake. All the more spectacular seems to me the finding that an increase of the rate of mass transfer perpendicular to the concentration direction and thus perpendicular to the direction of the molecular flux can lead to an increase of this very flux. In a possibly over-simplified picture, I take it as an indication that the probability of molecular propagation (i.e. of window passages) in flux direction is not uniform and varies, e.g., with the concentration in the cages into which the molecules are going to jump. In such a scenario, an enhancement of diffusion perpendicular to the flux direction would indeed be understood to, possibly, as well enhance the jump rate in flux direction, i.e. to give rise to flux enhancement. This would, moreover, nicely correlate with the diffusivity enhancement with increasing loading (Fig. 4).

I guess, after the revision the present paper is well suited for publication in Nature Communications. Leaving it to the authors to possibly refer to also the present considerations, I warmly recommend acceptance.

Response: We are glad to know that the reviewer appreciates the experimental findings, novelty of this work and recommends its publication.

Reviewer 3:

Although this manuscript has been improved according to the previous reviewing, still I do not think it can reach the level of Nature Communications. The novelty of this work, as you say, is the realization of diffusion controlling factor in heterogeneous pore system and chemical tuning of those factors. In my opinion, it is a little exaggeration. I do not think the current research results can support this novelty. Honestly, this is a nice try and some interesting results are presented. But Nature Communications is a top journal and only publishes high-quality research. Current work should be submitted to other specialized journal.

Response: We would like to respond to this criticism based on following points:

- i) Apart from simulation, there are no straightforward techniques which can give insight of the molecular diffusion direction in a porous solid. We have shown that this is possible, if we can make a model porous system.

- ii) The method of study, and the insight, i.e. how diffusion process is regulated by concentration gradient and pore window size is unprecedented.
- iii) The findings have a direct implication towards developing efficient membranes.

Based on these points, we believe that the findings are novel enough and suitable for Nature Communication.

The absolute values of free energy are meaningless, please use the relative values in Figure 5 to avoid the confusion.

Response: These are relative values, compared to position A and B.

Regarding to your opinion about the effect of MOF flexibility, I cannot agree it completely. In Figures S14 and S15, only one guest molecule is presented, as is not the real case. It means that you did not consider the concentration of guests inside MOF pores. It is not rigorous to say “there is no steric barrier caused by the side groups.....”. Maybe this effect of MOF flexibility will not alter current conclusion, but this kind of effect is indeed existed.

Response: We agree to this point, and that’s why we have already mentioned in the manuscript that “Note that flexibility (local and global) of the MOF structures can also influence diffusivity, and this aspect is not addressed in the current hypothesis.” We have highlighted this in yellow.

In conclusion, Nature Communications is an excellent journal and this manuscript is at this point not of the necessary caliber.

Response: We agree that Nature Communication is an excellent journal.

We do not agree that the work is not of the necessary caliber. The basis of the novelty is mentioned above.